# First Dose of the BNT162b2 mRNA COVID-19 Vaccine Reduces Symptom Duration and Viral Clearance in Healthcare Workers

**DOI:** 10.3390/vaccines9060659

**Published:** 2021-06-17

**Authors:** Luca Coppeta, Ottavia Balbi, Zaira Grattagliano, Grazia Genga Mina, Antonio Pietroiusti, Andrea Magrini, Matteo Bolcato, Marco Trabucco Aurilio

**Affiliations:** 1Department of Occupational Medicine, University of Rome Tor Vergata, 00188 Roma, Italy; lcoppeta@gmail.com (L.C.); ottaventidue@hotmail.it (O.B.); zaira.grattagliano33@gmail.com (Z.G.); graziagmina@gmail.com (G.G.M.); pietroiu@uniroma2.it (A.P.); andrea.magrini@uniroma2.it (A.M.); 2Legal Medicine, University of Padua, 35121 Padova, Italy; 3Department of Medicine and Health Sciences “Vincenzo Tiberio”, University of Molise, 86100 Campobasso, Italy; marco.trabuccoaurilio@unimol.it

**Keywords:** SARS-CoV-2, healthcare workers, COVID-19, vaccine, mRNA vaccine, first dose administration

## Abstract

Severe acute respiratory syndrome coronavirus 2 (SARS-CoV-2) has infected more than one hundred million people since the beginning of the worldwide pandemic. In this study, data from a large hospital in central Italy was used to evaluate the impact of the first dose of the BNT162b2 mRNA vaccine on SARS-CoV-2 infections in terms of the prevalence of symptomatic cases, symptom duration, and viral clearance timing. All vaccinated Healthcare Workers (HCWs) with positive RT-PCR by nasopharyngeal (NP) swabs were divided into two cohorts (positive RT-PCR within day 12 and positive RT-PCR between day 13 and day 21 after first dose administration) and compared for the presence and duration of symptoms and the timing of viral clearance. The same variables were evaluated across HCWs with positive RT-PCR within 6 days after first dose administration and non-vaccinated HCWs with positive RT-PCR between 1 October 2020 and 28 February 2021. Eighteen HCWs tested positive on RT-PCR by NP swab from day 1 to day 12 after the 1st dose administration (incidence rate 6.2 × 10^−4^) and 5 HCWs from day 13 to day 21 (incidence rate 2.3 × 10^−4^). Symptom duration and viral clearance timing are significantly shorter in the cohort of HCWs with positive RT-PCR 12 days after the first dose of the BNT162b2 mRNA vaccine. The administration of the first dose proved effective in reducing presence, symptom duration, and viral clearance even in HCWs vaccinated for less than 6 days. These results could have implications on public health and post-exposure prophylaxis.

## 1. Introduction

Since the beginning of the spread of SARS-CoV-2, more than one hundred million people have been infected [1], and the World Health Organization (WHO) declared the COVID-19 outbreak a pandemic on 11 March 2020 [2].

Although young people are also affected by the COVID-19 pandemic [3], the elderly, those with underlying medical conditions, and Health Care Workers (HCWs) are at the greatest risk of infection [4].

The COVID-19 pandemic has created serious consequences for public health systems in addition to substantial social and economic repercussions. In order to face this critical situation, stakeholders on the local, national, and global levels have coordinated efforts to develop safe and effective vaccines in order to contain COVID-19 [5].

As of 20 January 2021, at least seven different vaccines had been rolled out throughout various countries. At the same time, more than 200 additional vaccine candidates are in development, of which more than 60 are in clinical development [6].

On 21 December 2020, the European Medicine Agency (EMA) issued a positive assessment of safety and efficacy for the BNT162b2 mRNA vaccine, developed by BioNTech and Pfizer (Comirnaty), and, as a result, the European Commission granted conditional marketing authorization for the first vaccine against COVID-19 in Europe [7]. The BNT162b2 mRNA vaccine is a lipid nanoparticle–formulated, nucleoside-modified RNA that encodes the SARS-CoV-2 spike glycoprotein, modified by two proline mutations to lock it in the prefusion conformation [8]. The BNT162b2 mRNA vaccine is administered intramuscularly in two 30-μg doses, 21 days apart [9]. The clinical trial data is encouraging [8,10], and there is some evidence that even a single dose of BNT162b2 mRNA vaccine is protective against asymptomatic SARS-CoV-2 infection [11], but more studies are needed regarding the immunogenicity and efficacy of the first dose.

In line with the epidemiology of the disease and the evidence of the risk of severe disease and exposure to COVID-19, the Italian government identified those with underlying medical conditions and HCWs as priority groups for vaccination [12].

On 27 December 2020, *Vaccine day*, a vaccination campaign started across Europe, including Italy. On 28 December 2020, Policlinico Tor Vergata (PTV) in Rome started administering the first dose of the BNT162b2 mRNA vaccine to HCWs.

The aim of this study is to evaluate the impact of the first dose of BNT162b2 mRNA vaccine on SARS-CoV-2 infections in terms of the prevalence of symptomatic cases, symptom duration (days), and viral clearance timing (days between first positive RT-PCR and first negative RT-PCR by nasopharyngeal (NP) swab) in a population of occupationally exposed HCWs.

## 2. Materials and Methods

The study is a retrospective population-based cohort study on routinely collected data. After providing informed consent, HCWs of PTV received a dose of 0.3 mL of vaccine into the left or right deltoid. The dose was delivered using high-precision syringes after dilution of the vaccine with a sodium chloride 0.9% solution.

NP swabs were collected in accordance with recently published technical guidance [13].

In PTV’s accredited routine diagnostics laboratory, the Seegene Allplex 2019-nCoV assay was performed for the qualitative detection of SARS-CoV-2 viral nucleic acids in the swab specimens. The Seegene AllplexTM2019-nCov assay is an in vitro diagnostic (IVD) real-time reverse-transcriptase polymerase-chain reaction (RT-PCR) and is used to test NP swabs in line with the manufacturer’s protocols. The assay identifies the virus by multiplex real-time PCR targeting three viral genes (gene E, gene N, and gene RdRP), thus complying with internationally validated testing protocols. The “Seegene viewer” shows whether the exported data are 2019-nCoV Detected, Presumptive positive, or Negative for easy retrieval of the result by the user.

### 2.1. Study Population

The total HCWs working in PTV was 2956: 1553 MDs (50.9%), 912 nurses (31.9%), and 491 other Health Care Professionals (17.2%). Personal data and screening data with RT-PCR by NP swabs were collected for all HCWs newly vaccinated between 28 December 2020, and 28 February 2021.

Study outcomes included documented infection with the SARS-CoV-2 (positive RT-PCR by NP swab), presence of symptoms (HCWs were considered symptomatic when one or more of the following symptoms were present: fever or chills, cough, shortness of breath or difficulty breathing, fatigue, muscle or body aches, headache, new loss of taste or smell, sore throat, congestion or runny nose, nausea or vomiting, diarrhea), symptom duration and viral clearance timing calculated in days.

Since the most recent literature showed the early onset of a partially protective effect of immunization after the first dose with a divergence of cumulative incidence of COVID-19 cases over time among placebo and vaccine recipients was within 12 days of receiving the first dose [8], all vaccinated HCWs with positive RT-PCR by NP swabs were divided in two cohorts:

HCWs with positive RT-PCR by NP swab within day 12 after first dose of the BNT162b2 mRNA vaccine;HCWs with positive RT-PCR by NP swab between day 13 and day 21 after the first dose of the BNT162b2 mRNA vaccine.

We recorded the presence of symptoms immediately before or after the test, symptom duration, and viral clearance timing and then compared data concerning the presence and duration of symptoms and viral clearance timing of the two cohorts.

The subgroup of vaccinated HCWs with positive RT-PCR by NP swab within 6 days after first dose administration was compared, using the same variables, with non-vaccinated HCWs with positive RT-PCR by NP swab between 1 October 2020 and 28 February 2021. (Descriptive characteristics in Table 1).

### 2.2. Statistical analysis

Statistical analysis was performed using SPSS (version 26) (SPSS Inc., Chicago, IL, USA). The level of significance was set at a *p*-value < 0.05.

We calculated the incidence rates of SARS-CoV-2 infections in the group of HCWs which received the first dose and compared the incidence rate of the first period (from day one to day 12) with the second period before the booster vaccine dose (from day 13 to day 21).

In order to assess the effect of the first vaccine dose on symptom duration and viral clearance timing, we compared the data from the two cohorts by means of the ANOVA test for repeated measures, identifying sources of variation in continuous data.

For the comparison of categorical variables of the cohorts (presence of symptoms) we conducted the Chi-square test, whereas for the continuous ones (symptom duration and clearance), we used the Student’s *t*-test.

In order to estimate the consistency of vaccine efficacy in workers with high-risk exposure to SARS-CoV-2 close to the first dose administration, we also used a *t*-test for independent samples comparing data of HCWs with positive RT-PCR by NP swab within 6 days after the vaccine administration with those of non-vaccinated HCWs with positive RT-PCR by NP swab between 1 October 2020 and 28 February 2021.

## 3. Results

Two thousand and ninety-three first doses of the BNT162b2 mRNA vaccine were administered to PTV HCWs between 28 December 2020 and 28 February 2021, accounting for 80.9% of the HCWs. Two thousand and twenty-five booster doses were administered in the same period of time.

Ninety-one point six percent of MDs, 75.4% of nurses, and 57.4% of other HCWs (lab technicians, radiology technicians, perfusionist technicians, neurophysiopathology technicians, cardiocirculatory physiopathology technicians, psychologists, biologists, dieticians, dental hygienists, speech therapists, orthoptists, pharmacists) were vaccinated with the first dose.

The mean age was 39.64 ± 11.35 years, and 61.9% were female.

During the 21 days after the administration of the first vaccine dose, 23 HCWs tested positive on RT-PCR by NP swab: 18 HCWs from day 1 to day 12 (incidence rate 6.2 × 10^−4^) and 5 HCWs from day 13 to day 21 (incidence rate 2.3 × 10^−4^). The HCWs tested positive on RT-PCR from day 1 to day 12 and from day 13 to day 21 presented symptoms in 9/18 (50.0%) and 2/5 (40.0%) cases, respectively.

Descriptive characteristics of the study population by vaccination status and timing of newly positive RT-PCR tests are reported in Table 1.

As reported in Table 2, symptom duration and viral clearance timing were significantly shorter in the cohort of HCWs with positive RT-PCR by NP swab between day 13 and day 21 after the first dose of the BNT162b2 mRNA vaccine (*p* < 0.05).

Fifty-six point five two of cases (13/23) tested positive on RT-PCR by NP swab during the first week following administration. (Table 1). The administration of the first vaccine dose is effective in reducing the presence and duration of symptoms, and viral clearance timing even in HCWs vaccinated for less than 6 days; HCWs with positive RT-PCR by NP swab within the first 6 days after administration showed a significant reduction in the presence of symptoms and significantly shorter symptom duration and viral clearance timing compared to HCWs infected before the vaccine campaign began (Table 3).

## 4. Discussion

Most European countries, including Italy, are facing a shortage of vaccines. In this situation, all potential second-dose deferral strategies are considered in order to give even partial protection to the highest percentage of the population [11]. This strategy has not been studied and could expose people to incomplete protection for a long time. Such considerations are especially important when programming vaccine campaigns for workers exposed to high risk [14,15].

Observational studies considering events (infections) between the first and second dose, such as this one, can be used to estimate the effectiveness of a single dose without any impact on the timing of administration. Recent randomized studies of phase III demonstrated that the first dose of the BNT162b2 mRNA vaccine has an effectiveness of 52% (95% C.I. 29.5 68.4) 12 days after administration [10]. These results are based on the general population, which also includes non-working age subjects or people with no social interaction. This study suggests a high impact of one dose of the BNT162b2 mRNA vaccine on the incidence rate of infection, consistent with previously published data [16,17]. It also shows an impact in terms of mitigation of symptoms and reduction in viral timing between day 13 and day 21 after first dose administration. It is reasonable to assume that shorter symptom duration and shorter viral clearance timing have an impact on preventing transmission, with significant public health implications. Since the vaccine not only shortens viral clearance timing but might also decrease the amount of virus clearance, it could reduce transmission. If the vaccine reduces the potential for transmission, the public health benefits will be much greater [18].

Conversely, symptom mitigation and shorter viral clearance timings can lead to underdiagnosis of infection in asymptomatic and paucisymptomatic subjects, who are a significant vehicle of transmission of SARS-CoV-2 [19]. It is important to assess the potential implications of a shift towards more asymptomatic SARS-CoV-2 infections. The analysis conducted on the cohort of HCWs with positive RT-PCR by NP swab by day 6 after administration of the first dose showed a reduction in viral clearance timing and symptom duration. Since the median incubation time of SARS-CoV-2 is 5–6 days, this group of HCWs was probably infected immediately before or after vaccination [20,21]. We suppose that exposure occurred between 5 days before the administration of the first dose and 4 days after. Assuming the ineffectiveness of the first dose of vaccine for effective post-exposure prophylaxis, at this moment, the Advisory Committee on Immunization Practices (ACIP) recommends postponing the administration of the first dose of the BNT162b2 mRNA vaccine in subjects with high-risk exposure [22]. Nevertheless, the significant reduction in symptoms and the shorter viral clearance timing shown in this study suggest that even the administration of a vaccine dose shortly before high-risk exposure to SARS-CoV-2 could have a positive impact on the clinical course of infection and on public health. The high percentage of HCWs with positive RT-PCR by NP swab within the first week after administration could be due to misbehavior and inattention [23].

### Limits of the Study

The design of the study does contain limits since we could not evaluate the effectiveness of a single dose of the BNT162b2 mRNA vaccine for more than 21 days because all HCWs received the second dose on day 21. For the same reasons and due to the short study period in question, we did not analyze and compare the effectiveness of the second dose. The epidemiological trend between 1 October 2020 and 28 February 2021 was variable and difficult to compare with the trend in January and February 2021. Nevertheless, at present, there is no evidence that epidemiological trends and variants of SARS-CoV-2 have a negative impact on the presence and duration of symptoms and on viral clearance timing. In addition, the statistical value of this analysis is limited because of the low number of positive NP swabs collected between January and February 2021.

The time of testing (RT-PCR by NP swab) of HCWs in PTV is 15 days, and since it is below the mean viral clearance timing found in our study, this timeframe did not enable us to identify a significant percentage of asymptomatic subjects with positive RT-PCR by NP swab. The actual number of infected HCWs in our population could be higher, and vaccination efficacy could be overestimated.

## 5. Conclusions

This study showed that 12 days after administration, a single dose of the BNT162b2 mRNA vaccine is effective in reducing cases of infection in a population of occupationally exposed HCWs. A single dose of the BNT162b2 mRNA vaccine also significantly reduced the presence and duration of symptoms and viral clearance timing in HCWs with positive RT-PCR by NP swab. A single dose of the BNT162b2 mRNA vaccine also significantly reduced symptom duration and viral clearance timing in HCWs infected by SARS-CoV-2 immediately before or after administration of the first vaccine dose. This evidence could have implications on public health in terms of equity of access to vaccination [24] and post-exposure prophylaxis.

## Figures and Tables

**Table 1 vaccines-09-00659-t001:** Study population characteristics by vaccination status and timing of newly positive RT-PCR tests by NP swab.

Variables	HCWs Received 1st Dose of Vaccine Number of Subjects = 2393	Positive before 1st Dose Administration (1 October 2020–28 February 2021) Number of Subjects = 277	Positive after 1st Dose Administration Number of Subjects = 23
Sex			
Male	910 (38.1%)	97 (35.0%)	10 (43.5%)
Female	1483 (61.9%)	180 (65.0%)	13 (56.5%)
Age in years (mean, SD)	39.64 ± 11.35	41.30 ± 10.29	33.00 ± 8.69
HCWs			
MDs	1423 (91.6%)	115 (41.5%)	8 (34.8%)
Nurses	688 (75.4%)	126 (45.5%)	12 (52.2%)
Other HCWs *	282 (57.4%)	36 (13%)	3 (13.0%)
Timing (newly positive after 1st dose)			
Within 6 days	-	-	13 (56.52%)
Within 12 days	-	-	18 (78.26%)
13–21 days	-	-	5 (21.74%)

* Other HCWs: lab technicians, radiology technicians, perfusionist technicians, neurophysiopathology technicians, cardiocirculatory physiopathology technicians, psychologists, biologists, dieticians, dental hygienists, speech therapists, orthoptists, pharmacists.

**Table 2 vaccines-09-00659-t002:** Incidence of COVID-19 among vaccinated HCWs within 21 days of first dose administration.

Time Since 1st Dose	HCWs Received a First Dose of Vaccine *	Vaccinated HCWs Newly Positive for SARS-CoV-2	Incidence of COVID-19 among Vaccinated HCWs	Mean Duration of Viral Clearance	Mean Symptom Duration
	*n*	*n*	N/10.000 HCWs	Days ± SD	Days ± SD
1–12 days	2393	18	6.2	17.39 ± 4.828	3.00 ± 3.678
13–21 days	2375	5	2.3	10.20 ± 3.564	2.20 ± 3.194

* Incidence was calculated among HCWs without documented previous SARS-CoV-2 infection.

**Table 3 vaccines-09-00659-t003:** HCWs with positive RT-PCR by NP swab within 6 days after the vaccine administration compared with those of non-vaccinated HCWs with positive RT-PCR by NP swab between 1 October 2020 and 28 February 2021.

Variables	HCWs Newly Positive for SARS-CoV-2	Symptomatic HCWs Newly Positive for SARS-CoV-2	*p*-Value	Mean Duration of Viral Clearance	*p*-Value	Mean Symptom Duration	*p*-Value
	*n*	*n*		Days ± SD		Days ± SD	
Positive within 6 days after 1st dose	13	5	<0.01	16.15 ± 4.930	<0.05	2.38 ± 3.525	<0.05
Positive non-vaccinated (1 October 2020–28 February 2021)	277	200		23.16 ± 10.207		6.86 ± 6.470	

## Data Availability

The data presented in this study are available by request from the corresponding author. The data are not publicly available for ethical reasons.

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
