# Peer review of "First Dose of the BNT162b2 mRNA COVID-19 Vaccine Reduces Symptom Duration and Viral Clearance in Healthcare Workers"

_vaccines, 2021, doi:10.3390/vaccines9060659_

Round 1

Reviewer 1 Report

The clinical studies described in this manuscript are very interesting. It is a very important finding that a certain degree of SARS-CoV-2 infection prevention and symptom reduction effect is observed within 6 days after a single vaccination.

Reviewers would like to ask the following questions. Please discuss or provide data on those questions.

1. Reviewers cannot require to do more samples it because the authors use cohort data from more than 2000 people in their study, but the reviewer worries that the number of infections after vaccination is low. In fact, the proportion of infected people who developed symptoms is being compared between those who have been vaccinated within 6 days and those who have not been vaccinated. Is this really a statistically correct comparison and a significant difference even though in the case of  only 13 or 5 people versus more than 200 people? This should be explained in detail in the results and discussion.

2. The rationale for dividing the vaccinated group within 12 days and 13-21 days after vaccination is unknown. As a result of statistical analysis, the data claimed by the combination came out, so did you divide it so? Or is it for some other reason?

3. In relation to the question 2, is 12 days after vaccination an important turning point in post-vaccination immune induction? Or can we find the meaning of the turning point from the immunological findings of the COVID-19 vaccine? The above should be considered in detail.

Author Response

.... Reviewers cannot require to do more samples it because the authors use cohort data from more then 2000 people in their study, but the reviewer worries that the number of infections after vaccination is low. In fact, the proportion of infected people who developed symptomps is being compared between those who have been vaccinated within 6 days and those who have not been vaccinated. Is this a really statistically correct comparison and a significant difference even though in the case of only 13 or 5 people versus more than 200 people? This should be explained in detail in the results and discussion.

A: Thank you for your detailed consideration. We added in the limits of the study (lines 206-207) the fact that statistic value of this analysis is limited because of the little number of positive NP swabs collected between January and February 2021.

The rationale for dividing the vaccinated group within 12 days and 13-21 days after vaccination is unknown. As a result of statistical analysis, the data claimed by the combination come out, so did you divide it so? Or is it for some other reason?

In relation to the question 2, is 12 days after vaccination an important turning point in post-vaccination immune induction? Or can we find the meaning of the turning point form the immunological findings of the covid-19 vaccine? The above should be considered in detail.

A: Thank the reviewer for bringing up these two points. We added in lines 99-101 the rationale and the literature reference for dividing the vaccinated group within 12 days and 13-21 days after first dose.

We look forward to hearing from you regarding our submission. We would be glad to respond to any further questions and comments that you may have.

Reviewer 2 Report

In this retrospective analysis, Coppeta et.al summarize their finding on the protective effect of first/one-dose of BNT162b2 mRNA vaccine against COVID19 among HCW in Italy. The authors have analyzed the data from December 28, 2020, until February 28, 2021.  Overall, the information contained in this paper would be helpful. However, a significant weakness of this paper is the language use, sentence formation, and grammatical errors. This paper requires serious editing from a native English-speaking person.

General comments:

1). In the introduction, it would be helpful to mention about the nature of the vaccine (Comirnaty), since it would help to understand why a single dose of vaccine was thought to be sufficient.

2). It would be more meaningful and valuable to include the same analysis after the second dose of the Comirnaty vaccine.

Specific comments:

Line 69. Change “obtained” into “delivered.”

Line 93. It is unclear where is the data for these two “other cohorts”. Are these different cohorts? Where are their demographic information?

Table 2, the commas after numbers should be replaced with dots. “Nr”. Should be expanded as the “number of subjects” or “n”

In Table-2, it is unclear if the 2,375 cases at 13-21 days’ period are included in the 2,393 cases in the 1-12 days’ group? Or are they two different cohorts?

Author Response

We thank the reviewers for their comments and recommendations. The document has been reviewed by a native English speaker, we apologize because the changes are not highlighted as it has been extensively revised. We have addressed the issues raised as follows:.

1) In the introduction, it would be helpful to mention about the nature of the vaccine (Comirnaty), since it would help to understand why a single dose of vaccine was thought to be sufficient.

A: You make a fair assessment. In Lines 52-55 we added a brief description about the nature of the vaccine.

2) It would be more meaningful and valuable to include the same analysis after the second dose of the Comirnaty vaccine.

A: Thank you for your suggestion, we added it to the limits of the study (lines 220-221). The aim of this paper is to analyse data after the first dose, we will discuss further data in future studies.

3) Line 69. Change “obtained” into “delivered.”

A: Done

4) Line 93. It is unclear where is the data for these two “other cohorts”. Are these different cohorts? Where are their demographic information?

A: Thank you for your consideration. We have described (lines 113-116). two “other cohorts” in a different way: HCWs resulted positive within 6 days from the first dose administration (N=13) are a subgroup of total number of HCWs resulted positive between first and second dose (N=23), as mentioned on Table 1. This subgroup was compared with HCWs resulted positive before any dose administration, from October 1st 2020 to February 28th 2021 (N=277), population described on Table 1

5) Table 2, the commas after numbers should be replaced with dots. “Nr”. Should be expanded as the “number of subjects” or “n”.

A: Done.

6) In Table-2, it is unclear if the 2,375 cases at 13-21 days’ period are included in the 2,393 cases in the 1-12 days’ group? Or are they two different cohorts?

A: Thank the reviewer for bringing up this point. Incidence was calculated among HCWs without documented previous SARS-CoV-2 infection (we have now specified it on caption): 2,375 cases are HCWs susceptible to newly SARS-CoV-2 infection at 13-21 days, as known as 2,393 vaccinated HCWs deducted of 18 newly positive at 1-12 days from first dose administration.

Reviewer 3 Report

The authors described that the 1st dose of BNT162b2 reduced duration of symptoms and virus shedding of SARS-CoV-2, suggesting post-exposition prophylaxis. Although this finding is interesting and merits for publication, the English language should be improved. Especially, the wording on virus shedding or clearance need careful editing. 

Specific points:

  1. The authors should unify the expression of vaccine. They included BNT162b2 in the title but Comirnaty in the text.  
  2. The authors should indicate brief procedure of RT-PCR. Was quantitative assay carried out?
  3. Severity of index for symptomatic infection should be compared between cohorts.
  4. Lines 193-195. Since the vaccine not only shortened period of virus shedding but also might decrease amount of shedding virus, it could not increase transmissions. 

Author Response

We thank the reviewers for their comments and recommendations. The document has been reviewed by a native English speaker, we apologize because the changes are not highlighted as it has been extensively revised. We have addressed the issues raised as follows:.

... the wording on virus shedding or clearance need careful editing.

A: done

- The authors should unify the expression of vaccine. They included BNT162b2 in the title but Comirnaty in the text.  

A: Done.

- The authors should indicate brief procedure of RT-PCR. Was quantitative assay carried out?

A: Thank you for this suggestion. In Lines 76-85 we added a reference about the collecting technique of NP swabs and a description of the assay performed.

- Severity of index for symptomatic infection should be compared between cohorts.

A: All symptomatic infection were mild, no severe infection occurred in vaccinated population

- Lines 193-195. Since the vaccine not only shortened period of virus shedding but also might decrease amount of shedding virus, it could not increase transmissions. 

A: Agreed. We deleted the sentence and we strengthen the concept that vaccine could not increase transmission lines 195-196.

Round 2

Reviewer 2 Report

The authors' responses to my comments are satisfactory.  This manuscript may now be considered for publication.

Author Response

we want to thank the auditor for his work and attention

Reviewer 3 Report

I have no serious comments. A few minor revisions are listed below.

  1. Line 31. Delete HCWs.
  2. Line 53. Delete trimerized and RDB.
  3. Line 97. COVID-19 not Covid-19.
  4. Lines 214 and 215. Delete clearance. 

Author Response

we want to thank the auditor for his work and attention.

Line 31. Delete HCWs

A) Deleted.

Line 53. Delete trimerized and RDB.

A) Deleted.

Line 97. COVID-19 not Covid-19.

A) Done.

Lines 214 and 215. Delete clearance.

A) Deleted.